# Repurposing tRNAs for nonsense suppression

Suki Albers[1], Bertrand Beckert[1], Marco C. Matthies[2], Chandra Sekhar Mandava[3], Raphael Schuster[4], Carolin Seuring [5], Maria Riedner[4], Suparna Sanyal [3], Andrew E. Torda [2], Daniel N. Wilson [1] & Zoya Ignatova [1✉]

Three stop codons (UAA, UAG and UGA) terminate protein synthesis and are almost exclusively recognized by release factors. Here, we design de novo transfer RNAs (tRNAs) that efficiently decode UGA stop codons in *Escherichia coli*. The tRNA designs harness various functionally conserved aspects of sense-codon decoding tRNAs. Optimization within the TΨC-stem to stabilize binding to the elongation factor, displays the most potent effect in enhancing suppression activity. We determine the structure of the ribosome in a complex with the designed tRNA bound to a UGA stop codon in the A site at 2.9 Å resolution. In the context of the suppressor tRNA, the conformation of the UGA codon resembles that of a sense-codon rather than when canonical translation termination release factors are bound, suggesting conformational flexibility of the stop codons dependent on the nature of the A-site ligand. The systematic analysis, combined with structural insights, provides a rationale for targeted repurposing of tRNAs to correct devastating nonsense mutations that introduce a premature stop codon.

[1] Institute of Biochemistry and Molecular Biology, University of Hamburg, Hamburg, Germany. [2] Center for Bioinformatics, University of Hamburg, Hamburg, Germany. [3] Department of Cell and Molecular Biology, Uppsala University, Uppsala, Sweden. [4] Institute of Organic Chemistry, University of Hamburg, Hamburg, Germany. [5] Center for Structural and Systems Biology, Hamburg, Germany. ✉email: zoya.ignatova@uni-hamburg.de

In nature, the 61 triplet codons encoding the 20 amino acids in proteins are decoded by, on average, 40–46 distinct transfer RNAs (tRNAs or isoacceptors) in bacteria, 41–55 cytosolic and 22 mitochondrial tRNAs in eukaryotes[1–3]. However, three codons, UAA, UAG, and UGA, are almost exclusively reserved for terminating translation across the three kingdoms of life[4], and are instead, decoded by proteins termed release factors (RFs). Two decoding RFs are present in bacteria, RF1 and RF2, whereas all three stop codons are recognized by one factor, eRF1/aRF1 in eukaryotes and archaea, respectively[5]. At a mechanistic level, the nucleophilic reactions in the ribosomal peptidyl-transferase center in the case of sense-codon recognition by an aminoacyl-tRNA and RF-mediated hydrolysis of peptidyl-tRNA are markedly different[6].

Nonsense mutations within protein-coding sequences convert a sense triplet into a stop codon, which in humans is connected to various devastating pathologies[7,8]. In a few species, the detrimental effects of pervasive nonsense mutations are kept low by suppressor tRNAs, which commonly arise by mutation in a tRNA's anticodon[9] to decode the newly arising stop codon. However, through the exchange of the anticodon triplet only few natural tRNAs can be repurposed into suppressor tRNAs[10–15], generating tRNAs with fairly modest effectivity in decoding stop codons and correcting nonsense mutations. Multiple rounds of random nucleotide mutagenesis in the tRNA body[16] and combinatorial changes of different tRNA segments[17] have been shown to improve suppression efficiency, implying that other tRNA elements, in addition to the anticodon, can modulate this efficiency. Yet, this effect cannot be rationalized and it is often specific to the particular optimized tRNA.

In natural tRNAs, other parts outside of the anticodon, such as the anticodon loop or the TΨC-stem that interacts with the elongation factor (EF-Tu in bacteria), can compensate for the chemical diversity of the amino acid moiety and codon–anticodon strength variations, respectively, to achieve an optimal and similar decoding accuracy for all tRNAs[18]. In optimizing suppressor tRNAs to incorporate noncanonical amino acids, the destabilizing effect of those usually bulky amino acids has been counteracted through random mutations in TΨC-stem[19]. However, it remains unclear whether this compensatory principle shaping the decoding optimality by noncanonical amino acids can be applied to repurpose tRNAs to effectively decode premature stop codons and incorporate natural amino acids.

In this work, we implement a convergent de novo design of tRNAs[Ala] to complement the natural tRNA set with a new isoacceptor, which are refactored to decode UGA stop codon in Escherichia coli, and address the following questions. Firstly, how can the evolutionarily selected signature of a decoding tRNA be repurposed to promote efficient decoding of a stop codon? Secondly, what is the mechanism by which a repurposed suppressor tRNA decodes stop codons? Harnessing functionally conserved aspects of sense-codon decoding tRNAs (e.g., the conserved identity elements for aminoacylation from tRNA[Ala] and residues crucial to maintain tRNA architecture from structural studies with tRNAs), we computationally design tRNAs that function on the ribosome to suppress UGA stop codons. Additional fine-tuning of different tRNA sequence parts reveals that the TΨC-stem, which increases the binding affinity for EF-Tu, displays the highest effect in enhancing suppression activity. We solved the structure of the ribosome in a complex with the designed tRNA in the A-site. The conformation of the UGA codon is structurally similar to that of a sense codon rather than when translation termination RFs are bound. Our systematic analysis of various tRNA segments to modulate suppression activity combined with structural evidence provides a rationale for efficient repurposing of tRNAs to decode stop codons.

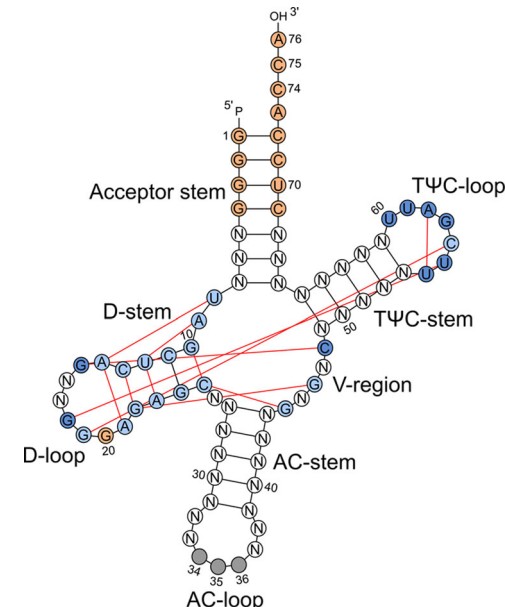

**Fig. 1 Nonsense suppressor tRNA design.** Fixed nucleotides in the design: AlaRS recognition (orange); anticodon (gray); tertiary interactions (red lines) between nts in t1 and t2 (dark and light blue) and in t3–t5 (light blue). N denotes sites optimized to fit to L-shaped tRNA structure. Different tRNA regions are designated. AC, anticodon.

## Results

**Design of nonsense suppressor tRNAs.** To accommodate within the same site on the ribosome, tRNAs share a common 3D architecture[20]. Yet, tRNA isoacceptors differ enough to be selectively charged with their cognate amino acids by the aminoacyl-tRNA synthetase (aaRS) and to serve their unique roles in decoding a specific codon[3]. Considering these functional rules (i.e., the identity elements for aminoacylation by the cognate aaRS) and structural constraints (i.e., maintaining interactions for establishing cloverleaf conformation and L-shaped tRNA architecture), we generated in silico different nonsense suppressor tRNAs (Fig. 1 and Table 1). For the design, we adopted the identity elements for aminoacylation from tRNA[Ala] by the cognate alanyl-tRNA synthetase (AlaRS). These identity elements are centered mainly within the acceptor stem (Fig. 1) and are independent of the sequence of the anticodon[21]. The conserved tertiary interactions were deduced from the crystal structure of the unmodified E. coli tRNA[Phe] (ref. [22]). The top five tRNA sequences, t1–t5 (Supplementary Table 1), were ranked by their folding probability to adopt cloverleaf secondary and L-shaped 3D structure. t1 and t2 maintain the maximal number of tertiary interactions as tRNA[Phe], whereas the tertiary interactions are reduced by ~30% for t3–t5 (Fig. 1).

**Designed tRNAs adopt translation-competent structures.** The t1–t5 tRNAs were synthesized in vitro with their CCA ends (that are genetically encoded in prokaryotes and posttranscriptionally added in eukaryotes[23]) and subjected to various tests to probe their integrity in translation. Correctly folded tRNAs have single-stranded NCCA-3′ termini that are crucial for aminoacylation. All five designed tRNAs folded into a structure compatible with single-stranded NCCA-3′ termini that are crucial for aminoacylation as revealed by probing, with a fluorescently labeled RNA/DNA hairpin oligonucleotide (Supplementary Fig. 1a) and, with the exception of the t2 variant, were aminoacylated (Fig. 2).

t1 displayed the highest aminoacylation level in vitro, comparable to that of both natural E. coli tRNA[Ala]GGC and

**Table 1 Design scheme of *E. coli* nonsense tRNA suppressors.**

| Properties | Sequence design |
|---|---|
| Secondary structure | (((((((..((((.......))))).(((((.......)))))…..(((((.......))))))))))))).... |
| Tertiary interactions | .......UAGCUCA**G**..**G**G.AGAGC..................G.G.**C**.....**UU**C**GAUU**............... |
| AlaRS identity | GGGG..............G.................................CUCCACCA |
| New anticodon | ...............................CUA........................................ |
| Combined restrains | GGGG...UAGCUCAG..GGGAGAGC........CUA.......G.G.C.....UUCGAUU........CUCCACCA |

Secondary structure and major restraints in the design are shown. Bold bases were considered in the designs t1 and t2 only.

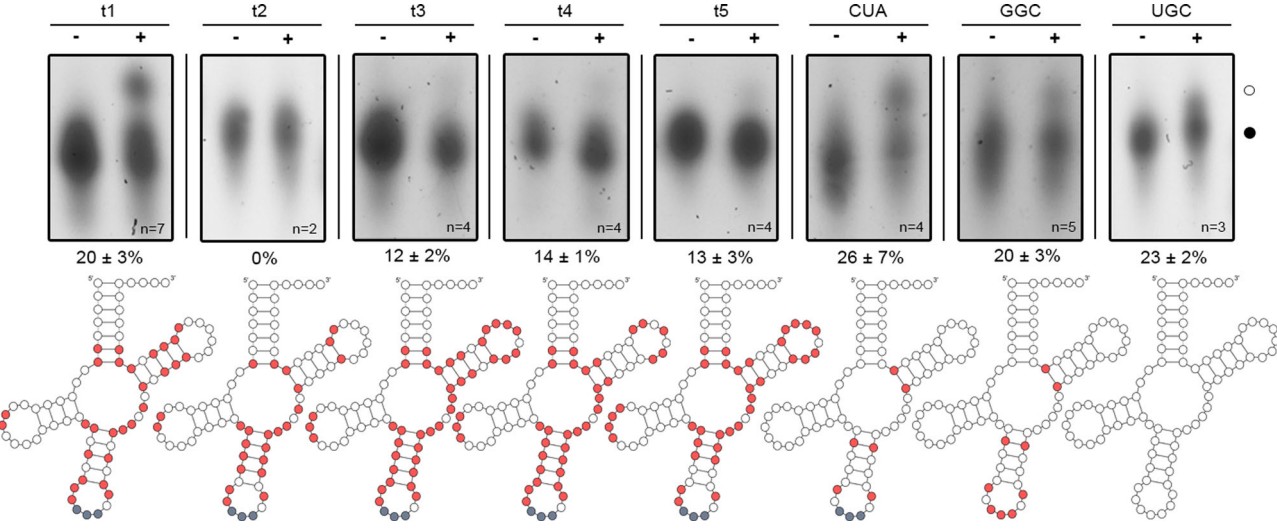

**Fig. 2 Designed nonsense suppressor tRNAs t1, t3–t5 are substrates of AlaRS.** Aminoacylation with Ala catalyzed by *E. coli* AlaRS (+) compared to nonaminoacylated in vitro transcribed tRNAs (−). Aminoacyl-tRNAs (○) migrate slower compared to non-acylated tRNAs (●). Aminoacylation levels are means ± s.d. (*n*, biologically independent experiments). In the schematic, nucleotides varied in each tRNA design, native tRNA$^{Ala}$GGC, and UAG-decoding tRNA$^{Ala}$CUA compared to the native tRNA$^{Ala}$UGC are highlighted in red; the anticodon CUA (gray) decoding UAG stop codon is the same for all engineered tRNAs. All replicates are provided as a Source data file.

tRNA$^{Ala}$UGC, as well as tRNA$^{Ala}$ with the anticodon exchanged to CUA so as to decode a UAG stop codon (Fig. 2). Although t2 bears the main identity element, a G3–U70 pair, for aminoacylation by AlaRS[21], no charging was observed (Fig. 2). An additional single base pair exchange at the end of the acceptor stem, C$_7$–G$_{66}$ to G$_7$–C$_{66}$, or in combination with G$_6$–C$_{67}$, rendering the acceptor stem identical to t1A3, restored the aminoacylation of t2 (Supplementary Fig. 1b), indicating the importance of proper stacking interactions within the acceptor stem for aminoacylation.

The in vivo decoding efficiency was tested by monitoring the ability of each designed tRNA to readthrough a UGA stop codon at position 29 (Ser29UGA) of green fluorescent protein (GFP). The *E. coli* XL1-blue expression strain, which is suitable for tRNA expression, contains a natural suppressor tRNA, *supE44*, which reads UAG stop codons (Supplementary Fig. 2). Thus, we changed the anticodon of the designed tRNAs to UCA to pair with the UGA stop codon. Despite comparable aminoacylation level as the native tRNAs$^{Ala}$, even the t1 variant with the highest charging level exhibited poor stop codon suppression (Fig. 3c).

**Tuning TΨC- and D-regions markedly enhances stop codon suppression.** The anticodon and its flanking regions within the anticodon stem and loop are crucial for accuracy during decoding[24–26]. Therefore, we systematically optimized the t1 anticodon loop taking into account various evolutionarily

conserved features[1] to enhance the accuracy of decoding. Among the anticodon loop alterations of t1A*i* designs (Fig. 3a), we included (i) the C$_{31}$–G$_{39}$ pair (t1A3–t1A6) ensuring a stable closure of the anticodon loop, (ii) A$_{37}$ (t1A4–t1A6) since it is conserved among natural tRNAs decoding codons beginning with U (ref. [27]), or (iii) introduced the U$_{32}$–A$_{38}$ (t1A5) pair interacting with nucleotide A1913 of the 23S rRNA, which has been shown to be crucial in tuning the efficiency of tRNA binding at the ribosomal decoding center[25]. These changes in the anticodon loop did not affect the aminoacylation levels of the t1A*i* designs, which remained comparable to t1 (Supplementary Fig. 4); however, we detected only marginal improvement of the efficiency to suppress UGA stop codons (Fig. 3c and Supplementary Fig. 3a), suggesting that the anticodon loop is not a major determinant for nonsense suppression.

Next, we focused on the TΨC-stem (Fig. 1), the sequence of which modulates EF-Tu affinity and thermodynamically compensates for the chemical diversity of the amino acid[28,29]. Using t1A3 as a template, we integrated TΨC-stem base pairs from the natural tRNA$^{Ala}$UGC, generating t1A3T1. Since alanine destabilizes the interactions with EF-Tu[18,28], to counteract the destabilizing effect of the amino acid we integrated the TΨC-stem from tRNA$^{Glu}$, which is one of the tRNAs with the strongest EF-Tu binding affinity, generating t1A3T2 (Fig. 3b). The stop codon suppression markedly improved with t1A3T2 in vivo (Fig. 3c, d and Supplementary Fig. 5), while t1A3T1 exhibited a suppression efficiency similar to that of t1A3 (Fig. 3c), suggesting

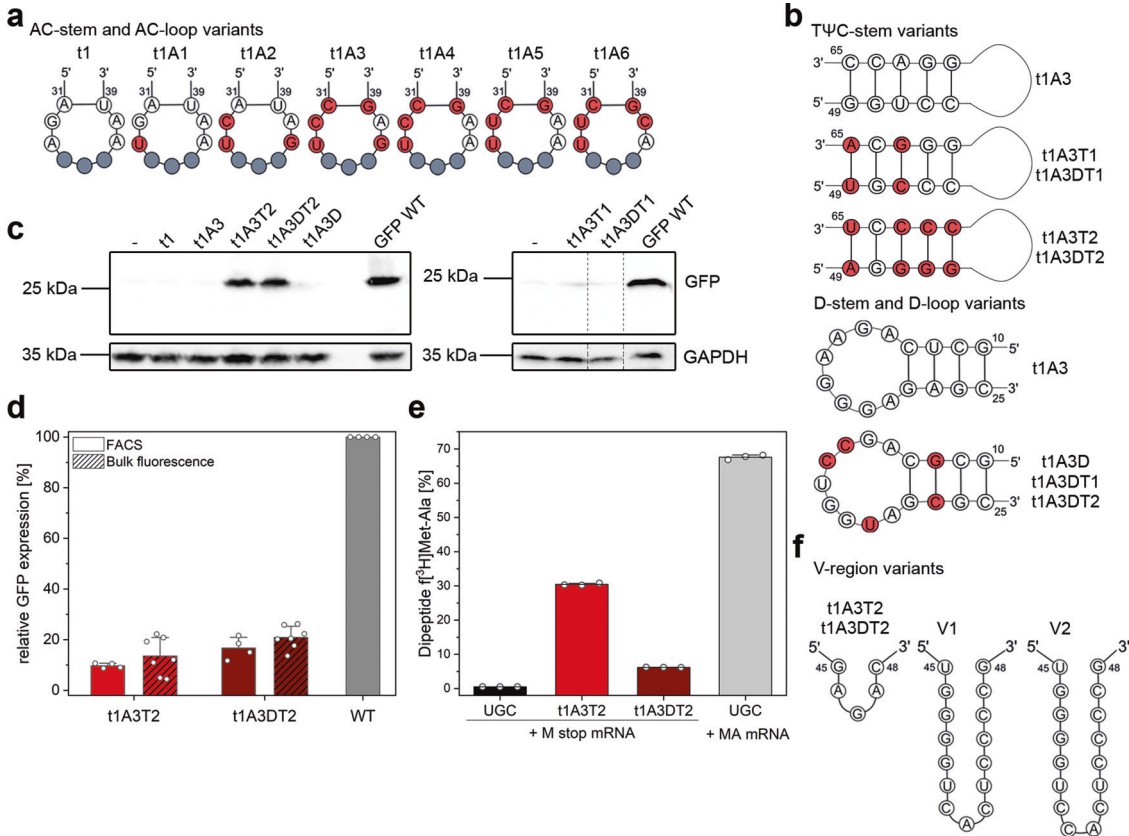

**Fig. 3 Engineered nonsense suppressor tRNAs are translationally competent. a** Sequence editing within the anticodon stem and loop of t1. Nucleotide substitutions highlighted in red; the anticodon UCA (gray) decoding UGA stop codon is the same for all engineered tRNAs. **b** Sequence editing within the TΨC-stem or D-stem and D-loop of t1A3. Nucleotide substitutions highlighted in red. **c, d** In vivo suppression efficiency of t1, anticodon-edited t1A3, TΨC-stem-edited t1A3T1 and t1A3T2 (light red), D-region-edited t1A3D, and TΨC- and D-region-edited t1A3DT1 and t1A3DT2 (dark red) tested in *E. coli* expressing an UGA-containing GFP variant detected by immunoblotting (**c**; $n = 3$–9 biologically independent experiments −, mock transformation), or flow cytometry and bulk fluorescence (**d**). Bulk fluorescence and flow cytometry signals were normalized to the expression of wild-type GFP (gray) whose expression was set to 100%. Data are means ± s.d for flow cytometry ($n = 4$ biologically independent experiments; Supplementary Fig. 5) and bulk fluorescence ($n = 7$ biologically independent experiments). Dashed vertical lanes (**c**) denote the place of excision of lanes with samples unrelated to this experiment. **e** Efficiency of in vitro transcribed tRNAs in dipeptide formation in a fully reconstituted *E. coli* translation system measured by the amount of f[³H] Met-Ala dipeptide using either Met-Ala (MA) or Met-stop(UGA) (M stop) mRNAs as the template. UGC (gray) is tRNA$^{Ala}$UGC. Data are means ± s.d. ($n = 3$ biologically independent experiments). **f** Variable loop extension of t1A3T2 and t1A3DT2. Source data to **c–e** are provided as a Source data file.

that strong EF-Tu interactions indeed enhance UGA decoding by the engineered tRNAs. t1A3T2 incorporated predominantly Ala at the UGA stop codon (Supplementary Fig. 6).

The D-stem is the binding platform of elongation factor P to stabilize tRNAs, which carry amino acids with slow peptide bond formation, such as tRNA$^{Pro}$ (ref. [30]) in the ribosomal P-site[31,32]. Thus, we integrated the specific signature of the D-stem and D-loop of tRNA$^{Pro}$ into t1A3, t1A3T1, or t1A3T2, generating t1A3D, t1A3DT1, and t1A3DT2, respectively (Fig. 3b). Thereby, all tertiary interactions except the A9–U12–A23 base triplet were maintained (Fig. 1). The D-stem and D-loop alone (t1A3D), or combined with the TΨC-stem base pairs of tRNA$^{Ala}$UGC (t1A3DT1), only marginally supported UGA suppression to a level comparable to that of t1 and t1A3 (Fig. 3c). The TΨC-stem with the tRNA$^{Ala}$ (T1) or tRNA$^{Glu}$ signature (T2) in combination with the D-stem and D-loop of tRNA$^{Pro}$, yielding t1A3DT1 or t1A3DT2, respectively, enhanced the suppression of the latter tRNA variant in vivo (Fig. 3c, d and Supplementary Fig. 5). However, the contribution of the D-stem/D-loop seems marginal, since the suppression activity of t1A3DT1 remained unchanged (Fig. 3c) and the higher mean value of t1A3DT2 than that of t1A3T2 was deemed statistically insignificant (Fig. 3d and Supplementary Fig. 5). Both t1A3T2 and t1A3DT2 formed f

[³H]Met-Ala dipeptide on Met-stop(UGA) mRNA in a fully reconstituted in vitro translation system, albeit with different efficiencies than in vivo, 31% and 6%, respectively (Fig. 3e and Supplementary Fig. 7). The control, wild-type tRNA$^{Ala}$UGC, did not form any dipeptide on that mRNA although it formed a saturating amount (~70%) of f[³H]Met-Ala dipeptide on the sense-codon containing Met-Ala(GCA) mRNA.

To further elucidate whether the in vivo and in vitro activity differences t1A3T2 and t1A3DT2 are variant specific, we also probed the effect of TΨC-stem and D-stem, as well as D-loop changes on another tRNA, t2AS2 (Supplementary Fig. 1b). We chose t2AS2 since t2 variants have a different acceptor stem than t1 (Fig. 2), and with the modification $C_7–G_{66}$ to $G_7–C_{66}$ the aminoacylation level of t2AS2 reached the level of t1 or t1A3 (Supplementary Fig. 1b). Similar to t1A3 variants, exchanging the TΨC-stem with the tRNA$^{Glu}$ signature (T2) potently enhanced suppression activity of t2AS2A3T2, while changing the D-stem/ D-loop of tRNA$^{Pro}$ alone only marginally enhanced the suppression of t2AS2A3D (Supplementary Fig. 3b). However, the combination of the D-stem/D-loop with the TΨC-stem (t2AS2A3DT2) decreased the suppression efficiency (Supplementary Fig. 3b, c), resembling the lower efficiency of t1A3DT2 in the in vitro fMet-Ala dipeptide formation assays (Fig. 3e).

Importantly, the TΨC-stem improves the suppression activity in synergy with the anticodon loop (compare t2AS2A3 with t2AS2T2 with t2AS2A3T2, Supplementary Fig. 3b).

The natural UGA decoder tRNA[Sec] possesses a long variable (V)-region[33]. Long V-regions are associated with higher nonsense suppression activity in mammals[34]. Consequently, we increased the V-region length of t1A3T2 and t1A3DT2 to be similar to that of mammalian suppressor tRNAs (Fig. 3f). Any alteration in the V-region decreased suppression levels to nearly that observed with t1 (Supplementary Fig. 3d).

Together, these data imply that fine-tuning of the TΨC-stem sequence majorly improves the stop codon suppression efficiency and together with the anticodon loop constitute two key regions to modulate, in order to repurpose tRNAs for efficient nonsense suppression. In turn, the D-region exhibits an effect that likely depends on the tRNA sequence and can range from marginally positive or neutral to negative, and counteract the effect of the TΨC-stem. It is also likely that the D-region generally decreases the suppression efficiency, but for some tRNA variants, posttranslational modifications in vivo may counteract the negative effect of the D-region rendering it slightly positive to neutral, as seen for the t1A3DT2 in vivo and in in vitro assay (Fig. 3d, e and Supplementary Fig. 5).

**Structural insights into the nonsense suppression mechanism by repurposed tRNAs.** To elucidate the structural basis for how t1A3T2 decodes a stop codon, we set out to determine a structure of t1A3T2 decoding a UGA stop codon on the ribosome. We employed an in vitro translation disome approach that exploits the erythromycin-dependent ErmCL leader peptide-mediated stalling of the ribosomes[35], to generate E. coli 70S ribosomes stalled at UGA stop codon in the ribosomal A-site (Supplementary Fig. 8a). The resulting 70S–UGA complex was incubated with affinity-purified t1A3T2 (Supplementary Fig. 8b), as well as an N-aminopropyl derivative of negamycin (AP-Neg), a translation inhibitor that induces misreading and stop codon suppression[36–39]. Single-particle cryo-EM reconstruction coupled with in silico sorting yielded one major homogenous subpopulation of 70S ribosomes with peptidyl-tRNA in the P-site, and deacylated tRNA bound in the A- and E-sites (Supplementary Fig. 9a), which could be refined to an average resolution of 2.9 Å (Fig. 4a, Supplementary Fig. 9b–f, and Supplementary Table 2). The cryo-EM density for the nascent polypeptide chain and P-site tRNA (Supplementary Fig. 10a–c) was consistent with previous ErmCL-ribosome structures[35], where erythromycin stalled translation of ErmCL such that the ErmCL-peptidyl-tRNA[Ile] located in the P-site, thereby precisely positioning the UGA stop codon in the A-site for our mRNA. In the A-site, the cryo-EM density for t1A3T2 suppressor tRNA was well-resolved (Fig. 4a and Supplementary Fig. 9d–f), enabling a molecular model to be built de novo (Fig. 4b). In particular, the $_{34}$UCA$_{36}$ anticodon of the t1A3T2 suppressor was observed to form perfect Watson–Crick base pairing with the UGA stop codon of the mRNA (i.e., A36 with U, C35 with G in the first and U34 with A; Fig. 4b). An overlay with the X-ray structure of wild-type tRNA[Ala]GGC decoding the cognate sense-codon GCC[25] suggests that the decoding by our nonsense suppressor tRNA is mechanistically identical to the decoding of sense codons by elongator aminoacyl-tRNAs (Fig. 4c). Indeed, all the hallmarks of interactions with cognate tRNA were observed, including defined conformations for the decoding site nucleotides G530, A1492, and A1493 (Supplementary Fig. 10d), an intact 32–38 base pair in the t1A3T2-tRNA, as well as an engaged interaction of A1913 with the t1A3T2-tRNA (Supplementary Fig. 10e, f). The decoding mechanism by t1A3T2 appears similar to that of tRNA[Sec]

(Fig. 4d), which is a natural isoacceptor recoding UGA stop codons to incorporate selenocysteine into proteins in conjunction with a downstream mRNA loop and assisted by specialized translation factor SelB in E. coli[40]. By contrast, the conformation of the UGA codon in the context of the t1A3T2 suppressor seen here differs from that observed during canonical translation termination by RF2 (refs. [41–43]; Fig. 4e). While the first two bases stack upon each other, they are not recognized by A1492 and A1493, and the third base is rotated away from the first two, where it stacks upon G530 (Fig. 4f). This suggests that the conformation of stop codons is not preformed, but rather forms upon binding of the A-site ligand, and that the distinct conformations adopted depend upon the nature of the A-site ligand, i.e., whether it is a tRNA or RF.

**AP-Negamycin stabilizes nonsense suppressor tRNAs in the A-site.** The structure also shows an additional density within the decoding center, directly adjacent to the anticodon stem loop of t1A3T2 (Fig. 5a, b), which we attributed to AP-Neg based on the similarity to the Neg-binding site[37,39]. AP-Neg forms two interactions with the t1A3T2 anticodon stem loop, namely, a direct interaction between the N4 secondary amine of AP-Neg and the non-bridging oxygen of the U$_{34}$ 5′ phosphate, as well as an indirect interaction between the terminal carboxyl group of AP-Neg and the non-bridging oxygen of the U$_{34}$ 3′ phosphate via a hydrated magnesium ion (Fig. 5c and Supplementary Fig. 10g–i). For AP-Neg, we do not observe previously reported binding in the ribosomal exit tunnel on the archaeal 50S subunit[44] or any of the eight additional binding sites observed on the Thermus thermophilus 70S ribosome[39]. Thus, we propose that AP-Neg facilitates nonsense suppression by stabilizing suppressor tRNA binding at the A-site, analogous to the mechanism proposed by which Neg promotes misreading of sense codons[37,39]. Our structure also rationalizes the improved biochemical and antimicrobial activities of AP-Neg over Neg[45], since the aminopropyl chain of AP-Neg forms additional contacts with the 16S rRNA. This includes potential hydrogen bonds from the terminal amino group with the 2′ OH of the ribose and non-bridging oxygen of the 3′ phosphate of A968 (Fig. 5d and Supplementary Fig. 10g–i). Consistently, substitutions of the amino group of AP-Neg generate Neg analogs with reduced inhibitory activity[45].

**Discussion**

In summary, we have designed tRNAs that function on the ribosome and efficiently suppress UGA stop codons in E. coli. The route for designing de novo nonsense suppressor tRNAs is based on initial computational design paired with sequence fine-tuning of functionally important tRNA elements. Our results show that several factors, including structurally important interactions, functional parameters for aminoacylation, and maintenance of decoding accuracy, need to be considered for efficient repurposing of tRNAs to decode stop codons. Thereby, optimization within the TΨC-stem, which modulates the binding affinity for EF-Tu, displayed the most potent effect in enhancing suppression activity. Base pairs 49–65, 50–64, and 51–63 of the TΨC-stem exhibit the strongest influence on EF-Tu binding:weak EF-Tu:aminoacyl-tRNA binding limits the formation of ternary complex; while strong binding reduces the dissociation rates from EF-Tu·GDP after cognate codon–anticodon interaction[18,28].

The structure of the t1A3T2 suppressor in the ribosomal A-site provides the structural basis in understanding the mechanism of stop codon suppression. The structural results suggest that the decoding mechanism of nonsense suppressor tRNAs is markedly different than the canonical stop codon recognition by RF1 and RF2, which naturally decode stop codons during translation[41–43].

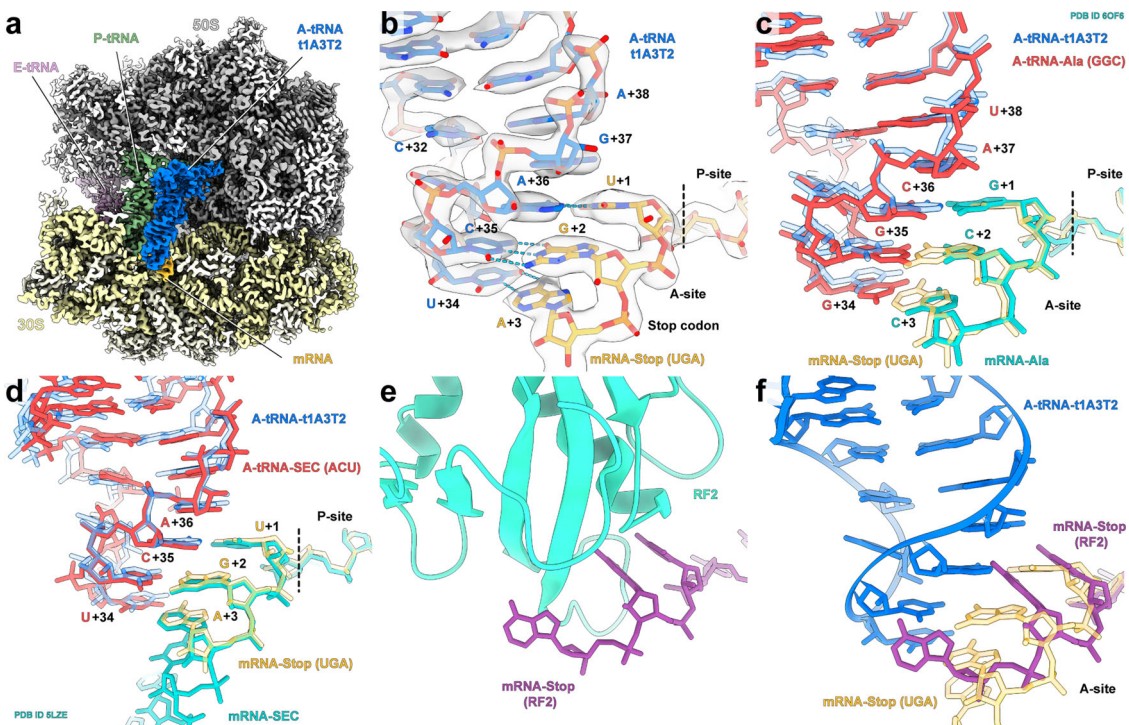

**Fig. 4 t1A3T2 decoding mechanism is identical to elongator tRNAs. a** Cryo-EM reconstruction of the t1A3T2 decoding UGA on the ribosome with segmented densities for t1A3T2 (blue), P tRNA (green), E tRNA (pink), 30S (yellow), and 50S (gray). **b** Close-up view of the anticodon stem loop of t1A3T2 (blue) showing three Watson–Crick pairing decoding UGA (pale orange). **c** Same view than **b** with an overlay between the cognate tRNA^AlaGGC (red) bound to GCC codon (turquoise; PDB ID 6OF6)[25]. **d** Same view than **b** with an overlay between the Sec-tRNA^Sec (red) bound to UGA codon (turquoise; PDB ID 5LZE)[40]. **e** Interaction of RF2 (cyan) with the decoding center (mRNA stop, violet; PDB ID 4V4T)[75]. **f** Same view than **e** without RF2 and overlay of t1A3T2 (blue) decoding UGA (pale orange).

By contrast, the stop codon suppression mechanism by repurposed tRNA is identical to the decoding of sense codons by elongator tRNAs and similar to the SelB-assisted tRNA^Sec decoding of UGA codon to incorporate selenocysteine[40]. However, tRNA^Sec inspired designs were inactive (Fig. 3f and Supplementary Fig. 3d), emphasizing the importance of the dedicated translation factor, tRNA^Sec, and the mRNA structural element as a suppression system rather than the tRNA alone. Finally, our results show plasticity of the stop codon in the decoding center such that the nature of the A-site ligand (i.e., tRNA or RF) can induce distinct stop codon conformations and thereby allow correction of nonsense mutations or expansion of the genetic code.

The chemical and sequential differences among tRNAs are compensated by the common 3D architecture, the unique set of consensus residues throughout tRNAs and the number of posttranscriptional modifications in the anticodon loop and the EF-Tu tertiary core, resulting in similar binding affinities to the ribosomal A-site and equivalent functions in decoding[18,46]. Systematic mutation of single-nucleotide pairs in the TΨC-stem reveals an additive contribution of each pair, thus enabling a reasonably accurate prediction of the EF-Tu binding affinity[47]. The thermodynamic contributions of nucleotide pairs to the binding of the eukaryotic elongation factor, eEF1A, have not yet been rigorously assessed. However, eEF1A and EF-Tu use the same homologous regions with conserved sites to bind aminoacyl-tRNAs[48]. Hence, it is conceivable that the optimization principles we established for repurposing bacterial tRNAs are transferrable to eukaryotic tRNAs. Supportive for this is the observation that among many eukaryotic tRNAs whose anticodon is exchanged to decode stop codons, tRNA^Leu exhibits very high efficiency[34]. Using the thermodynamic contributions of

single-nucleotide pairs of prokaryotic tRNAs (ref. [18] and the references therein), we calculated that the TΨC-stem of tRNA^Leu-eEF1A complex is among the ternary complexes with the highest putative stability. In addition, posttranscriptional modifications may also modulate the binding affinities in the eEF1A tertiary core. Three nucleotide pairs, 49–65, 50–64, and 51–63, from the TΨC-stem majorly contribute to the binding to elongation factor[47]. Among the human tRNAs, only nucleotides 49 and 50 are posttranscriptionally modified in ~¼ of the tRNAs (Modomics data base, http://genesilico.pl/modomics/), though the precise modification type and extent is by far incomplete.

Historically, the use of repurposed tRNAs to decode stop codons has been an attractive strategy proposed already to correct nonsense mutations causally linked to various diseases (reviewed in ref. [8]). Despite its enormous therapeutic potential, up to date no clinical trial has been launched. Most of the attempts to recode nonsense mutation-induced stop codon are based on repurposing of the anticodon, which yield suppressor tRNAs with modest activity in vivo[10–15]. Random mutagenesis-based improvements of activity[16] are also less attractive because they bear a tRNA-specific signature and independent random mutagenesis needs to be performed for any tRNA of interest. By contrast, our systematic analysis combined with structural evidence rationalizes the importance of specific tRNA segments, which can be targetably modified to modulate the suppression activity of any tRNA.

A rapidly developing area—the orthogonal translation—also employs a tRNA-suppression strategy to refactor stop codons and incorporate noncanonical amino acids to expand genetic code with new functionalities (reviewed in refs. [49–51]). Here, the amino acid substrates are unnatural, thus repurposing of the tRNA requires a more complex strategy involving modification of the

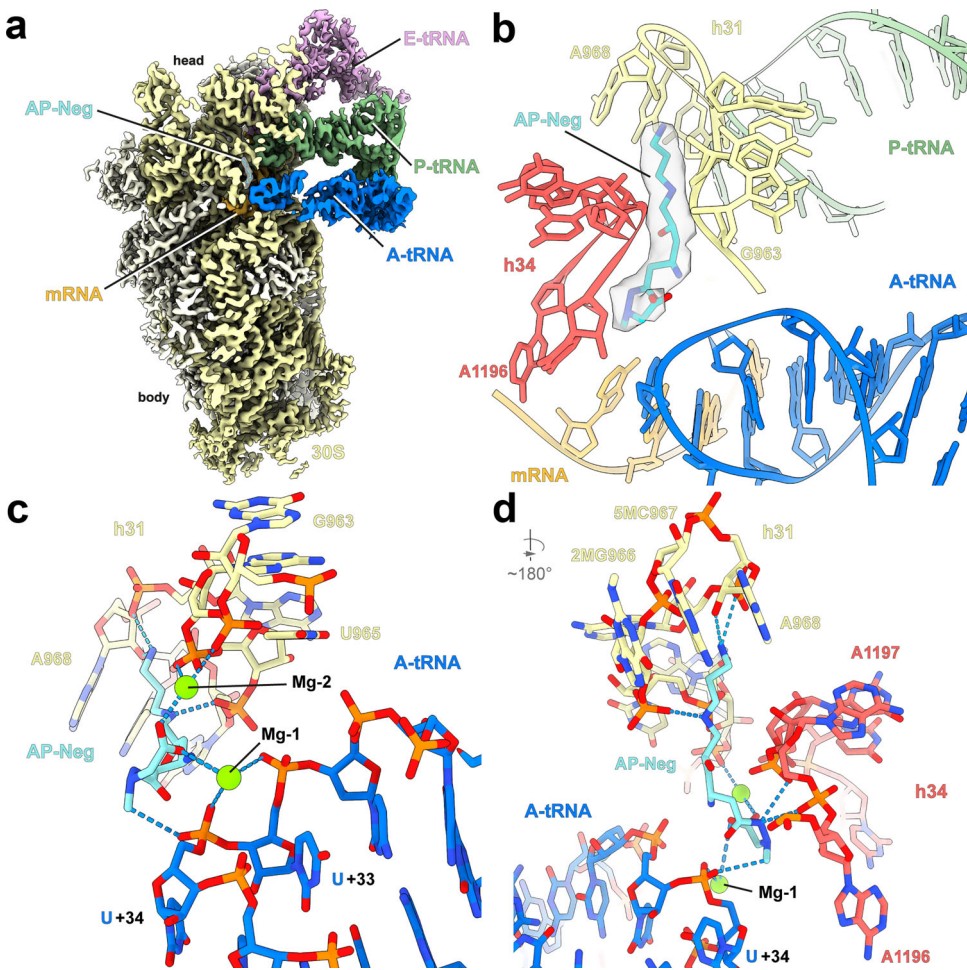

**Fig. 5 Decoding of t1A3T2-tRNA on the ribosome in the presence of AP-Neg. a** Cryo-EM reconstruction of the t1A3T2 decoding UGA on the ribosome with segmented densities for t1A3T2 (blue), P tRNA (green), E tRNA (pink), 30 S (yellow), and AP-Negamycin (transparent). **b–d** Close-up view of AP-Neg (cyan) bound to the 30S subunit (helix 34 and 31) in close proximity to the A-site tRNA (blue).

recognition elements for the orthogonal aaRS and acceptor stem[19,52–56]. The region we identified as the most crucial for enhancing the stop codon decoding activity with natural amino acids, the TΨC-stem, is also a potent modulator of the efficiency of tRNA-based incorporation noncanonical amino acids[19,52,54]. Since the noncanonical amino acids are on average bulkier than the natural amino acids, mutations in the TΨC-stem that would stabilize the interactions with EF-Tu and counteract the amino acid destabilization appears to be a successful strategy for further enhancing the decoding abilities of the repurposed tRNAs, independent of whether they incorporate natural or noncanonical amino acids.

## Methods

**Plasmids, expression, and growth curves**. tRNA sequences were cloned into pBST NAV2 (kindly provided by Dr. Axel Innis, Institut Européen de Chimie et Biologie, France) under control of a consecutive *lpp* promotor and *rrnC* terminator.

Wild-type GFP was cloned into pBAD33 under control of the L-arabinose-inducible promoter $P_{BAD}$. To obtain the GFP stop variants, the serine AGC codon at position 29 was substituted by either UAA, UAG, or UGA. For purification of the GFP variants for mass spectrometry, all constructs bear a C-terminal 6×His-tag.

All plasmids were expressed in XL1-blue cells at 37 °C in LB medium supplemented with ampicillin (100 µg/mL) and/or chloramphenicol (34 µg/mL) as selection marker.

**tRNA design**. Sequences were designed using DSS-Opt[57] with default parameters. Conserved tertiary interactions were deduced from the crystal structure of unmodified *E. coli* tRNA$^{Phe}$ (PDB ID: 3L0U)[22] and set as a maximal boundary of number of tertiary interactions. Sites conserved amongst *E. coli* tRNAs$^{Ala}$ for

charging with aaRS were also restrained (Table 1)[21]. Combining these restraints, for each design, 10,000 sequences were computationally designed and ranked by the predicted probability of folding as calculated by the ViennaRNA package (version 2.3.4)[58]. Target secondary structure and tertiary interactions were calculated with DSSR[59] from the coordinates of PDB entry 3L0U[22]. The probability of each sequence to form a cloverleaf secondary structure (*P(S)*) is assessed based on the calculation of the full equilibrium partition function (formula (16) ref. [60]). The tertiary interactions consisted of base triples (U8, A14, A21), (A9, U12, A23), (G10, C25, G44), (C13, G22, G46); base pairs (G15, C48), (G18, U55), (G19, C56), (U54, A58); and calculated hydrogen bonds in which bases C11, G24, G57, U59, and U60 participate (Table 1). G15, G18, C48, U54, U55, G57, A58, U59, and U60 were considered in the designs t1 and t2 only. Numbering is according to the *E. coli* tRNA$^{Phe}$ (PDB ID 3L0U)[22]. The top five ranked tRNA sequences, t1–t5, were selected (Supplementary Table 1) and used in functional assays.

**tRNA synthesis**. tRNAs were transcribed in vitro using T7 transcription system. Templates were generated by annealing of two partially overlapping DNA oligo-nucleotides bearing the tRNA sequence and an upstream T7 promoter (primers are listed in Supplementary Table 3). 24 µM of both overlapping oligonucleotides were denatured for 2 min at 95 °C and incubated for 3 min at room temperature in 20 mM Tris-HCl (pH 7.5). Primer extension was performed using 0.4 mM dNTPs and 4 U/µL RevertAid Reverse Transcriptase (Thermo Fisher Scientific) for 40 min at 37 °C. The dsDNA template was extracted with phenol/chloroform, ethanol precipitated, washed with 80% EtOH, and resuspended in DEPC-H$_2$O.

In vitro T7 transcription of the dsDNA template was performed in the presence of 2 mM NTPs, 1.25–5 mM GMP, 1× transcription buffer, and 0.6 U/µL T7 RNA polymerase (Thermo Fisher Scientific) overnight at 37 °C. The reactions were ethanol precipitated and purified by preparative denaturing polyacrylamide gel electrophoresis (PAGE). tRNAs were eluted in 50 mM KOAc and 200 mM KCl pH 7.0 overnight at 4 °C by rotating at 1000 r.p.m. The eluted tRNAs were filtered to remove gel pieces, ethanol precipitated, washed with 80% EtOH, and resuspended in DEPC-H$_2$O. tRNA integrity was monitored by denaturing PAGE.

**3′-CCA termini integrity of the synthesized tRNAs**. The integrity of the single-stranded 3′-CCA ends of the in vitro transcribed tRNAs was tested with a fluorescently labeled RNA/DNA stem loop oligonucleotide (Supplementary Table 3) ligated to the tRNA with 2.5 U T4 DNA ligase (Thermo Fisher Scientific) in 5 μL T4 DNA ligase buffer with 15% (v/v) DMSO overnight at 16 °C. tRNAs with ligated hairpin oligonucleotide and intact CCA termini were separated from the bulk tRNAs on denaturing PAGE. RNAs were visualized by fluorescence or SYBR™ Gold Nucleic Acid Stain. The approach visualizes only tRNAs with intact CCA-3′ ends and isoacceptors with aberrantly in vitro synthesized 3′ ends or truncated CCA ends in vivo remain invisible[61].

**In vitro aminoacylation**. tRNA folding and in vitro aminoacylation reactions were performed as described[62] with 1 μM purified E. coli AlaRS (plasmid kindly provided by Ya-Ming Hou, Jefferson Univ, PA). Aminoacyl-tRNAs were precipitated with ethanol and directly dissolved in 2× acidic RNA loading dye (pH 4.5). Charged and uncharged tRNA fractions were separated by acidic denaturing PAGE (6.5% (19:1) acrylamide:bisacrylamide, 8 M urea, and 0.1 M NaOAc pH 5) at 4 °C. tRNAs were visualized by SYBR™ Gold Nucleic Acid Stain. The fraction of aminoacyl-tRNA was determined by the intensity of the aminoacyl-tRNA with decreased mobility normalized to the intensity of the initial tRNA amount (i.e., the sum of aminoacyl-tRNA and nonaminoacyl-tRNA bands). The close migration of the aminoacyl-tRNA and nonaminoacyl-tRNA bands makes it difficult to quantify; the approach is rather suited for semiquantitative assessments whereby many replicates should be included.

**GFP readthrough assays**. The pBST NAV2 bearing different tRNAs (Primers are listed in Supplementary Table 3) and pBAD33 encoding GFP UGA were cotransformed in E. coli XL1-blue cells, and grown in LB medium containing ampicillin (100 μg/mL) and chloramphenicol (34 μg/mL). At $OD_{600\,nm}$ of 0.4, GFP expression was induced with 0.05 or 0.25% L-arabinose and cells were further cultivated till $OD_{600\,nm}$ 1.0.

To determine natural readthrough in absence of plasmid-encoded tRNAs, pBAD33 bearing GFP UAA, UAG, or UGA (primers are listed in Supplementary Table 3) were transformed into XL1-blue cells, grown in LB medium containing chloramphenicol (34 μg/mL). Cells were induced at $OD_{600\,nm}$ 0.4 with 0.05% L-arabinose and cultured for 2 h. GFP expression was probed by immunoblotting using anti-GFP antibody (anti-GFP from mouse IgG1κ, 1:1000 dilution; Roche) and HRP-conjugated goat anti-mouse secondary antibody (Immunstar goat anti-mouse-HRP, 1:10,000 dilution, Bio-Rad). Membranes were stripped and thereafter, probed with a HRP-conjugated anti-GAPDH antibody (GAPDH loading control monoclonal antibody (GA1R), HRP, 1:1000 dilution, Thermo Fisher Scientific). Furthermore, GFP expression was measured in bulk (480 nm excitation/530 nm emission; black 96-well plates with clear bottom (Corning)) and by flow cytometry on FACS Calibur with CellQuest™ Pro Software Version 6.0 (Becton Dickinson), where the GFP fluorescence was recorded for a total of ~100,000 events with the following settings: FSC = E01, log, SSC = 400, log, FL1 = 736, log and the following threshold: FSC = 52. The median fluorescence intensity was quantified using FlowJo Version 10.7.2. Cells were gated using FSC and SSC parameters. Autofluorescence background was subtracted and the mean of the medians of the biological replicates was normalized to the wild-type GFP fluorescence.

**Mass spectrometry**. XL1-blue cells transformed with pBAD33 wild-type GFP-6×His or cotransformed with pBAD33 GFP UGA-6×His (primers are listed in Supplementary Table 3) and pBST NAV2 t1A3DT2-tRNA, were grown in LB medium containing the corresponding antibiotics, induced at $OD_{600\,nm}$ 0.4 with 0.25% L-arabinose and further cultivated for 4 h. Cells were harvested by centrifugation at $5000 \times g$ for 15 min at 4 °C, resuspended in buffer A (20 mM Tris-HCl pH 8.0 and 500 mM NaCl), and lysed by four repetitive cycles of cryogenic disruption for 2 min at a frequency of 300 1/s using RetschMill MM400 (Retsch). Cell lysates were incubated with Ni-NTA agarose (Thermo Fisher Scientific) overnight under agitation at 4 °C. Nonspecifically bound proteins were removed by washing with buffer A containing 0–50 mM imidazole, and His-tag containing GFP was eluted with buffer A containing 100–200 mM imidazole, concentrated with Microcon 30, separated by 12% SDS–PAGE and stained with Coomassie. A band corresponding in size to that of GFP was excised from the gel.

For trypsination, an in-gel digest was performed as described[63]. Briefly, gel bands of interest were sliced and shrinking and swelling was performed using ACN and 100 mM $NH_4HCO_3$. Samples were reduced with 10 mM DTT (in 100 mM $NH_4HCO_3$) for 30 min at 56 °C, alkylated with 55 mM IAA (in 100 mM $NH_4HCO_3$) for 20 min at room temperature in the dark, and digested with trypsin (20 ng/μL in 100 mM $NH_4HCO_3$, Carl Roth) overnight at 37 °C. Peptides from the tryptic digest were extracted with ACN/ddH₂O 65/35 v/v + 5% formic acid (FA) and lyophilized. Thereafter, they were dissolved in 0.1% FA and injected into an UHPLC-system (Dionex Ultimate 3000, Thermo Fisher Scientific) coupled to an ESI-Q-TOF mass spectrometer (maXis, Bruker Daltonik). Separation took place on a reversed-phase separation column (EC 100/3 NUCLEODUR C-18-Gravity-SB 3 μm, Macherey-Nagel) using following stepwise gradient with ddH₂O + 0.1% FA (buffer A) and ACN + 0.1% FA (buffer B) and a flow rate of 300 μL/min: 3–30% buffer B in 60 min followed by 30–70% buffer B in 20 min. Spectra were recorded in positive ion mode, with a spectra rate of 1.0 Hz and a mass range of 100–2700

Da. The capillary voltage was set to 4500 V, the nebulizer pressure to 3.0 bar, the drying gas flow to 8.0 L/min, and the source temperature to 200 °C. MS/MS spectra were recorded using collision-induced dissociation in data-dependent acquisition mode. The number of precursors was set to 3, the preferred charge state to 2–5, and the intensity threshold to 2890 counts. Data interpretation were performed using the Data Analysis software (Version 4.2, Bruker Daltonik).

**Dipeptide formation assay**. All translation components were from E. coli, expressed and purified as described earlier[64]. Two mixes, initiation mix (IM) and elongation mix (EM), were prepared in HEPES polymix buffer (pH 7.5) containing energy pump components GTP (1 mM), ATP (1 mM), phosphoenol pyruvate (10 mM), pyruvate kinase (50 μg/ml), and myokinase (2 μg/ml). The IM contained 70S ribosome (1 μM), XR7 mRNA (2 μM) encoding either Met-Ala-stop(UAA) (MA mRNA) or Met-stop(UGA) (M stop mRNA), f[3H]Met-tRNA^fMet (1 μM), and the initiation factors IF1 (1 μM), IF2 (2 μM), and IF3 (1 μM). The IM was incubated at 37 °C for 15 min to form a proper 70 S initiation complex. The EM contained EF-Tu (20 μM), EF-Ts (2.5 μM), in vitro transcribed tRNA variants (5 μM), alanine (0.2 mM), and AlaRS (0.5 μM). The EM was incubated at 37 °C for 60 min to ensure maximal charging of the tRNAs. Equal volumes of IM and EM were mixed manually and quenched with 50% HCOOH after incubation of 1 min for the wild-type tRNA^Ala for MA mRNA, and 90 min for all other tRNA–mRNA combinations. The extent of dipeptide f[3H]Met-Ala formed was determined by separating it from f[3H]Met on a reverse phase C-18 column (Merck) connected to HPLC (Waters Co.) with inline radio flow detector (Beta-RAM 6, Lab logic)[65].

**Affinity purification of tRNAs from E. coli cells**. XL1-blue cells transformed with pBST NAV2 t1A3T2-tRNA were grown in LB medium containing ampicillin (100 μg/mL) until $OD_{600\,nm}$ of 1.0. Total RNA was extracted using TRIzol according to the manufacturer's instructions (Ambion). A total of 5–50 μg of total RNA were hybridized to 1 μL (100 pmol) of a 5′-biotinylated DNA oligo complementary to the 3′-half of the tRNA t1A3T2 (Supplementary Table 3) in 100 μl of 5× SSC buffer (750 mM NaCl and 75 mM trisodium citrate). The sample was denatured for 3 min at 90 °C followed by incubation for 10 min at 65 °C. A total of 100 μL Strepatividin MicroBeads (Miltenyi Biotec) were added to the RNA–DNA hybrid and incubated for 30 min at room temperature in a wheel shaker. Beads were washed two times with nucleic acid equilibration buffer (Miltenyi Biotec) and two times with 5× SCC buffer. Bound tRNAs were eluted by adding four times 100 μL DEPC-H₂O heated to 80 °C. The DNA oligonucleotide was removed with DNase I (Thermo Fisher Scientific), tRNA was extracted using phenol/chloroform, precipitated with ethanol, washed with 80% ethanol, and resuspended in DEPC-H₂O. The tRNA, which by this isolation procedure is deacylated, was analyzed by denaturing PAGE. Total RNA from untransformed XL1-blue cells, which do not express tRNA t1A3T2, was also subjected to affinity purification and used as negative control.

**Generation and purification of 70S–UGA complex**. The ErmCL UGA stop codon construct and ribosome complexes were generated following the same procedure as described[35]. Briefly, under the control of the T7 promoter (italicized in the sequence below) two consecutive ErmCL ORFs (shaded gray) with strong ribosome-binding site (gray bold) 7 nt upstream of the ATG start codon were synthesized (Eurofins, Germany). Both ErmCL ORFs were separated by 22 nt linker enabling an independent initiation of both ORFs; a complementary DNA oligonucleotide required for RNase H cleavage was annealed to parts of this linker (underlined). Both ErmCL ORFs contained a UGA stop codon at position 10 (bold). The complete sequence of the 2× ErmCL UGA construct is: 5′-*TAATACGACTCACTATAG*GGAGGTTTTAT A**AGGAGG**AAAAAATATGGGCATTTTTAGTATTTTT GTAATC**TGA**ACAGT TCATTATCAACCAAACAAAAAATAAA**GTTTTATA**AGGAGG**AAAAAATAT GGGCATTTTTAGTATTTTTGTAATC**TGA**ACAGTTCATTATCAACCAAACA AAAAATAA-3′ In vitro coupled transcription and translation of the 2× ErmCL UGA construct was performed using the Rapid Translation System RTS 100 E. coli HY Kit (biotechrabbit) in presence or absence of 10 μM erythromycin for 1 h at 37 °C. The ErmCL-stalled disomes were isolated on 10–40% sucrose gradients in buffer A, containing 25 mM HEPES-KOH pH 7.5, 150 mM KOAc, 15 mM Mg(OAc)₂, 1 mM DTT, 0.01% n-dodecyl D-maltoside, and 50 μM erythromycin, by centrifugation at $34,380 \times g$ (SW-40 Ti, Beckman Coulter) overnight at 4 °C. The disome fraction was then pelleted by centrifugation at $109,760 \times g$ (Ti70.1, Beckman Coulter) overnight at 4 °C and resuspended in buffer B, containing 25 mM HEPES-KOH pH 7.5, 100 mM KOAc, and 25 mM Mg(OAc)₂. Conversion into monosomes was performed first by annealing of a complementary DNA oligonucleotide (5′-TTCCTCCTTATAAAACT-3′, Microsynth) for 5 min at 30 °C, directly followed by a cleavage of the RNA–DNA hybrid by RNase H (Thermo Fisher Scientific) at 25 °C for 1 h. The reaction was layered on 10–40% sucrose gradient in buffer A and the monosomes were fractionated by centrifugation at $34,380 \times g$ (SW-40 Ti, Beckman Coulter), directly followed by a pelleting centrifugation step at $109,760 \times g$ (Ti70.1, Beckman Coulter) overnight at 4 °C. The stalled monosomes were resuspended in buffer B to a final concentration of 150 A$_{260\,nm}$/mL.

**Cryo-EM and single-particle reconstruction**. The complex reconstitution was performed by mixing 6.75 A$_{260\,nm}$ units/mL of purified 70S–UGA complex with the nonaminoacylated nonsense suppressor tRNA t1A3T2 isolated from E. coli cells

and 100 μM AP-NEG. A total of 3 μL of the complex was then applied on carbon-coated Quantifoil R3/3 holey grids and vitrified using Vitrobot Mark IV (FEI). Data collection were performed using EPU (FEI) on a Talos Arctica operating at 200 KeV equipped with a direct detector Falcon III (Thermo Fisher Scientific) at a defocus range between 0.7 and 3 μm. With an average of 30 micrograph per hour, in total 600 micrographs were recorded in counting mode (with a dose of 1 e$^-$/Å$^2$) and divided into 40 frames with a pixel size of 0.96 Å. Dose-fractionated movies were aligned using MotionCor2 v1.2.1 and RELION 3.1 (ref. [66]) and determination of the CTF, defocus values, and astigmatism were performed using GCTF (version 1.06)[67]. Automatic particle picking was then performed using Gautomatch (version 0.56; http://www.mrclmb.cam.ac.uk/kzhang/) and single particles were processed using RELION 3.1. The initial set of picked particles (79,477) was first subjected to an extensive 2D classification resulting in 65,608 particles. 3D refinement was performed using *E. coli* 70S ribosome as a reference structure. The particles were then further 3D classified resulting in a major population of 53,115 particles. After a final 3D refinement, CTF refinement and Bayesian polishing, the reconstruction yielded an average resolution of 2.8 Å according to FSC$_{0.143}$ criterion. The final maps were sharpened by dividing the maps by the modulation transfer function of the detector and by applying an automatically determined negative *B* factor to the maps, using RELION 3.1. All the maps were filtered according to local resolution using SPHIRE (SPARX, version 4.0)[68].

**Molecular model and figures preparation**. The molecular model for the ribosomal proteins and rRNA core was based on the molecular model of the *E. coli* 70S ribosome (PDB ID 6H4N)[69]. The models were rigid-body fitted into the cryo-EM density map using UCSF Chimera[70] followed by refinement, using Coot (version 0.8 and 0.92)[71] and ISOLDE (version 1.1)[72]. For the nonsense t1A3T2 suppressor, the molecular model was performed based on the density and 2D structure prediction. ErmCL nascent chain was de novo modeled. Erythromycin and AP-Neg were de novo modeled using ChemDraw, as well as ELBOW (phenix). All atomic coordinates were refined using Phenix (dev-2947-000)[73].

Figures showing electron densities and atomic models were generated using either UCSF Chimera (version 1.14), ChimeraX (version 1.0)[74] or PyMol Molecular Graphic Systems (version 2.4, Schrödinger).

**Reporting summary**. Further information on research design is available in the Nature Research Reporting Summary linked to this article.

## Data availability
The cryo-EM map has been deposited in the Electron Microscopy Data Bank (EMDB) with accession codes EMD-12035. The molecular model has been deposited in the Protein Data Bank (PDB) with accession codes 7B5K. The code for RNA sequence design is available at github [https://github.com/marcom/dss-opt]. All other data are available in the main text or the Supplementary Materials. Source data are provided with this paper.

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

## Acknowledgements

We thank Cornelia Cazey and Kay Grünewald for support at the Cryo-EM facility at CSSB, Hamburg, and Greg Basarab (University of Cape Town) for providing the AP-Neg. This research was supported by grants from the Deutsche Forschungsgemeinschaft IG73/14-2 (to Z.I.) and WI2381/6-1 (to D.N.W.), and Swedish Research Council 2016-06264, 2018-05946, and 2018-05498 (to S.S.). The work at the Cryo-EM facility (CSSB) is supported by University of Hamburg and grant from the Deutsche For-schungsgemeinschaft INST 152/775-1 FUGG.

## Author contributions

S.A. and Z.I. designed the study. M.M. and A.E.T. designed computationally t1–t5 variants. S.A. designed all mutations in the t1–t5, and performed the biochemical and in-cell studies. B.B. and C.S. collected the cryo-EM data. B.B and D.N.W. processed the cryo-EM data, built, and refined the molecular models. C.S.M. and S.S. performed the bio-chemical analysis in in vitro translation system. R.S. and M.R. performed the mass spec-trometry analysis. All authors interpreted the results and helped S.A. and Z.I. write the paper.

## Funding

## Competing interests

The Regents of the University of Hamburg have patents issued and pending for tRNA technologies on which S.A., Z.I., M.C.M., and A.E.T. are inventors. The remaining authors declare no competing interests.
