## [Peer Review File · Nature Communications]

REVIEWER COMMENTS

Reviewer #1 (Remarks to the Author):

The study by Ignatova and colleagues addresses the important question of refactoring tRNA-Ala for decoding a stop codon. This is a big open question in the field of translation, because nonsense mutations that result in premature translation termination are pathological, however a suppressor tRNA can decode the corresponding newly formed stop codons. Yet, the mechanism of tRNA repurposing has not been thoroughly studied.

The authors generated different nonsense suppressor tRNAs, synthesised them with the CCA ends and assessed a compatibility for aminoacylation and translation in *E. coli*. A variant was selected and systematically optimised in the anticodon loop to enhance the accuracy of decoding. The authors found that only marginal improvement could be achieved, and therefore concluded that the anticodon loop is not a major factor determinant. Then, the sequence modulating EF-Tu activity was reengineered using base pairs from the natural tRNA-Ala, which resulted in a markedly improved suppression of the stop codon. Also, reengineering of the D-stem loop increased suppression.

Using cryo-EM, the structure of the 70S-UGA complex with the engineered tRNA has been determined to 2.9 Å resolution. This structure provides key mechanistic insights. While, it showed the characteristics of a cognate tRNA, the observed conformation of the UGA codon is different from the canonical translation termination step. In addition, the decoding center surprisingly was found to accommodate AP-Neg that suggests a novel mechanism of nonsense suppression by stabilising suppressor tRNA binding at the A-site.

Overall, the analysis is outstanding, it reveals important novel findings, and well presented. I also like the concise nature of the report.

Minor stylistic suggestions for the authors to consider:

- 1) Abstract: It read well, but perhaps mention the specific mechanistic details that represent the main findings of the study. I think it would make the abstract more effective.
- 2) page 2, "... average, 40 different tRNAs ...", I understand what the authors mean, but given that different branches of organisms are included here, also organelles, it might be difficult to state the exact average. Maybe better to indicate a range, from min to max?
- 3) page 5, "... excellent quality of the cryo-EM density ...", better to avoid subjective descriptions.
- 4) page 6, "... differs dramatically ...", also a subjective description.
- 5) page 6, "AP-Neg", please kindly provide a brief description of the terminology and what it means for a non-expert reader.
- 6) page 6, last paragraph, change "A site" to "A-site" for consistency.
- 7) page 7, is "high-resolution" a meaningful description?
- 8) Methods, I think "RELION" would be more accurate, as it is an abbreviation.
- 9) Supplementary Fig. 8: For clarity, is it possible to please have all the density maps and models in the same size.
- 10) Supplementary Fig. 8: It is now standard in the field to include the angular distribution plots, FSC model-map, classification scheme, local resolution view.

11) Supplementary Fig. 10: Please consider moving to the main, as the structural work shown here represents a substantial fraction of the results and described in the text in detail.

12) The table for data and model statistics is missing?

Reviewed by A. Amunts

Reviewer #2 (Remarks to the Author):

I have read this manuscript with a great deal of interest but was frustrated by the lack of genuine novelty, a somewhat obsolete approach, and an outdated introduction that ignored similar and already published studies. Hence, I cannot recommend this manuscript for publication in Nature Communications.

In brief, Suki Albers and colleagues mutated E. coli tRNA-Ala to engineer artificial tRNA variants that turn stop-codons into sense-codons in E. coli cells. The authors then used cryo-EM to describe how the engineered tRNAs bind to E. coli ribosomes, in the presence of antibiotic negamycin. The authors claim that their data can help combat human disorders that are caused by premature termination of protein synthesis. I find it far-fetched.

My decision to reject the manuscript stems primarily from the fact that the authors ignored most similar and already published studies:

1. Superficial introduction on tRNA suppressors:

The authors ignore key previous studies and the overall state of the art in the field of suppressor tRNAs, and their approach to the engineering of suppressor tRNAs appears to be fully trivial.

For instance, the authors omit nearly every milestone paper on suppressor tRNAs engineering/applications, including classical studies (e.g. <https://pubmed.ncbi.nlm.nih.gov/6803169/>) and more recent studies of the past two decades by Peter Schultz, Jason Chin, George Church, Michael Jewett, Abhishek Chatterjee and many other leaders in the field who have been routinely engineering hundreds of suppressor tRNAs to alter the rules of the genetic code to enable new research tools and help solve important problems in biotechnology and medicine.

2. Outdated view on tRNA/mRNA interactions:

What I found particularly frustrating is their statements about the novelty of their structural observations. The fact that suppressor tRNAs are recognized by the ribosome as any other tRNAs is extremely trivial and already known. For instance, Marina Rodnina and Holger Stark observed the very same fact in their cryo-EM structure of E. coli ribosome bound to the suppressor tRNA, tRNA-Sec (<https://pubmed.ncbi.nlm.nih.gov/27842381/>).

Having read their introduction, I was expecting to find their engineered tRNAs outperforming all currently existing suppressor tRNAs in genetically-defective human cell lines (e.g. cell lines from beta-thalassemia patients), a test that is commonly used in the field. Instead, I found that the authors used E. coli and existing technologies to confirm existing knowledge. I suggest journals like ACS Chemical Biology as a proper fit for this work.

Reviewer #3 (Remarks to the Author):

The manuscript by Albers and colleagues aims to use tRNA design principles to engineer an improved tRNA that can be used to suppress stop codons. If such a nonsense suppressor tRNA could be identified, it could be useful to suppress premature stop codons that cause human disease. To identify an improved nonsense suppressor tRNA, the authors use the well-defined *E. coli* translation system. They first choose tRNA(Ala) as a sequence framework, due to the fact that this tRNA has nucleotide identity elements for charging by alanyl-tRNA synthase (AlaRS) that are not located in or near the anticodon. They predominantly reside in the acceptor stem, with one additional nucleotide in the D loop (Fig. 1). The authors first generated 10,000 tRNA variants computationally and narrowed these down to 5 based on predicted secondary and tertiary folding algorithms. These tRNAs, t1-t5, were then transcribed in vitro and tested for aminoacylation. Of these five, t1 was the most efficiently aminoacylated (Fig. 2) and was used for further engineering to improve in vivo nonsense suppression of a GFP reporter. Mutations in t1 in the anticodon stem-loop and the variable arm were ineffective, but mutations that optimized the TphiC stem, with additional D loop modifications (tRNA t1A3DT2), improved nonsense suppression dramatically (Fig. 3C-D). These TphiC stem and D loop mutations were inspired by biochemical insights into how other tRNA bodies are tuned for translation. The authors chose tRNA(Glu) for TphiC mutations and tRNA(Pro) for D loop mutations, due to tight binding of tRNA(Glu) to EF-Tu and interactions of tRNA(Pro) with EF-P when in the P site. The authors further confirmed incorporation of Ala using tandem MS/MS analysis (Fig. S5). The authors then used purified 70S ribosomes loaded with tRNA t1A3T2 to determine a 2.9 Å cryo-EM structure of the ribosome with a stop codon and tRNA t1A3T2 in the A site. The structure shows that the tRNA has the expected conformational features of an elongator tRNA properly bound in the decoding site, with the previously defined rRNA-tRNA contacts.

Overall, this is an interesting paper that will be of wide interest to the translation field. However, there are a few critical holes in the presentation the authors should fill. These are described below.

Major

1. The authors used T7-transcribed tRNAs for all of their biochemical work in vitro. Yet presumably the tRNAs are at least post-transcriptionally modified to some extent in cells. The authors see differences between tRNA behavior in vitro compared to in vivo that could be explained by these differences in modification. For example, nonsense suppression in vivo for t1A3DT2 is much better than t1A3T2 in vivo (Fig. 3C-D), but is worse in the dipeptide assay (Fig. 3E). The authors should map the post-transcriptional modifications of t1A3T2 and t1A3DT2, at least.
2. The authors made acceptor stem changes to t2 (t2AS2 and t2AS3) that made these variants essentially as good as t1A3. (The authors also don't make clear that t2AS3 and t1 have identical acceptor stems, which confused the reader.) Yet they did not test their utility in the GFP nonsense suppression assay in cells. Could these work as well as the t1A3T2 and/or t1A3DT2 variants, i.e. would the TphiC stem loop and D loop variants be required for the t2AS3 family of tRNAs?
3. The in vitro aminoacylation experiments in Fig. 2 are somewhat confusing. The levels of charging for t1 and CUA seem reasonable, but the percentages for t3-t5, GGC and UGC seem much higher than supported by the gel shown. The gel is a bit smeary, which could lead to overestimation in these samples.
4. Related to the aminoacylation assays, the authors used a ligation approach to ensure the ACCA was present on the transcribed tRNAs (Fig. S1). However, the ligation is far from quantitative. Could the presence of N+1 (non-templated addition) species be the cause of lower aminoacylation efficiency in vitro? Did the authors run a sequencing-grade gel to assess the amount of N+1 tRNA in their preparations? Have the authors tried the approach of Kao et al. (1999) RNA 5, 1268-ff to reduce non-templated species?

5. The FACS bar graphs in Fig. 3C and Fig. S3B should be accompanied by the FACS histograms, to show the cutoff value for GFP fluorescence used for defining GFP levels.

6. Fig. S9. It is not at all obvious that tRNA(Ile) is enriched in this experiment. Is this plot correct? Perhaps simply using the quality of the cryo-EM density of the tRNA and mRNA in the P site, it would be possible to show the tRNA is consistent with tRNA(Ile)?

7. All of the work was done in *E. coli*, yet the motivation is in part to address human disease. I don't think it's necessary to do more experimental work in this regard, but I do think the Discussion should be expanded to address what's known and what's not known about tRNA functional interactions in the human context. This is especially true when considering the differences in EF-Tu vs. EF1-alpha, and the likely differences in tRNA modifications that would occur in human vs. *E. coli* cells.

Minor:

1. For comparison purposes, the authors should show the GFP levels from t1A3 in Fig. 3C and 3D, instead of just in Figure S3A.

2. The right half of Fig. S7A seems incorrect. Should this not show purified disomes collapsing to monosomes after RNaseH treatment?

3. The authors should state that the affinity-purified tRNA for cryo-EM was not aminoacylated when the ribosome complex was formed. This was not clear upon first reading.

Reviewer #1:

The study by Ignatova and colleagues addresses the important question of refactoring tRNA-Ala for decoding a stop codon. This is a big open question in the field of translation, because nonsense mutations that result in premature translation termination are pathological, however a suppressor tRNA can decode the corresponding newly formed stop codons. Yet, the mechanism of tRNA repurposing has not been thoroughly studied.

The authors generated different nonsense suppressor tRNAs, synthesised them with the CCA ends and assessed a compatibility for aminoacylation and translation in E. coli. A variant was selected and systematically optimised in the anticodon loop to enhance the accuracy of decoding. The authors found that only marginal improvement could be achieved, and therefore concluded that the anticodon loop is not a major factor determinant. Then, the sequence modulating EF-Tu activity was reengineered using base pairs from the natural tRNA-Ala, which resulted in a markedly improved suppression of the stop codon. Also, reengineering of the D-stem loop increased suppression.

Using cryo-EM, the structure of the 70S-UGA complex with the engineered tRNA has been determined to 2.9 Å resolution. This structure provides key mechanistic insights. While, it showed the characteristics of a cognate tRNA, the observed conformation of the UGA codon is different from the canonical translation termination step. In addition, the decoding center surprisingly was found to accommodate AP-Neg that suggests a novel mechanism of nonsense suppression by stabilising suppressor tRNA binding at the A-site.

Overall, the analysis is outstanding, it reveals important novel findings, and well presented. I also like the concise nature of the report. Reviewed by A. Amunts

We were pleased to read the overall assessment of Dr. Amunts' review and are thankful for the remarks he made.

Minor stylistic suggestions for the authors to consider:

1) Abstract: It read well, but perhaps mention the specific mechanistic details that represent the main findings of the study. I think it would make the abstract more effective.

We have edited the abstract to include some specific details on our main findings.

2) page 2, "... average, 40 different tRNAs ...", I understand what the authors mean, but given that different branches of organisms are included here, also organelles, it might be difficult to state the exact average. Maybe better to indicate a range, from min to max?

We have included this information on page 2.

3) page 5, "... excellent quality of the cryo-EM density ...", better to avoid subjective descriptions.

We have changed the sentence (page 6) to read "The cryo-EM density for the nascent polypeptide chain and P-site tRNA..."

4) page 6, "... differs dramatically ...", also a subjective description.

We removed "dramatically".

5) page 6, "AP-Neg", please kindly provide a brief description of the terminology and what it means for a non-expert reader.

The term AP-Neg is defined on page 6 as "an N-aminopropyl derivative of negamycin (AP-Neg) ..."

6) page 6, last paragraph, change “A site” to “A-site” for consistency.

Throughout the text we consistently hyphenate A-site only when it is linked with another noun e.g. A-site ligand. In all other occurrences, where it is itself a noun we use non-hyphenated A site. Consistently, we have corrected this also in the figures.

7) page 7, is “high-resolution” a meaningful description?

We have removed the term “high resolution”.

8) Methods, I think “RELION” would be more accurate, as it is an abbreviation.

We have changed “Relion” to “RELION”.

9) Supplementary Fig. 8: For clarity, is it possible to please have all the density maps and models in the same size.

All density maps and models are now shown the same size in Supplementary Fig 8.

10) Supplementary Fig. 8: It is now standard in the field to include the angular distribution plots, FSC model-map, classification scheme, local resolution view.

We have now included the angular distribution plot in Supplementary Fig. 8a. The FSC model-map is in Supplementary Fig. 8b. The classification scheme and local resolution view were present in panels Supplementary Fig. 8a and 8c, 8e, respectively.

11) Supplementary Fig. 10: Please consider moving to the main, as the structural work shown here represents a substantial fraction of the results and described in the text in detail.

We have reworked the main figures so that Figure 4 focuses now on the decoding aspect and the new Figure 5 focusses on the AP-negamycin. We thank the Reviewer for this suggestion.

12) The table for data and model statistics is missing?

We apologize for not having included the table of data and model statistics which is now included in the revised version as a new Supplementary Table 1.

Reviewer #2:

I have read this manuscript with a great deal of interest but was frustrated by the lack of genuine novelty, a somewhat obsolete approach, and an outdated introduction that ignored similar and already published studies. Hence, I cannot recommend this manuscript for publication in Nature Communications.

In brief, Suki Albers and colleagues mutated E. coli tRNA-Ala to engineer artificial tRNA variants that turn stop-codons into sense-codons in E. coli cells. The authors then used cryo-EM to describe how the engineered tRNAs bind to E. coli ribosomes, in the presence of antibiotic negamycin. The authors claim that their data can help combat human disorders that are caused by premature termination of protein synthesis. I find it far-fetched.

My decision to reject the manuscript stems primarily from the fact that the authors ignored most similar and already published studies:

1. Superficial introduction on tRNA suppressors:

The authors ignore key previous studies and the overall state of the art in the field of suppressor tRNAs, and their approach to the engineering of suppressor tRNAs appears to be fully trivial.

For instance, the authors omit nearly every milestone paper on suppressor tRNAs engineering/applications, including classical studies (e.g. <https://pubmed.ncbi.nlm.nih.gov/6803169/>) and more recent studies of the past two decades by Peter Schultz, Jason Chin, George Church, Michael Jewett, Abhishek Chatterjee and many other leaders in the field who have been routinely engineering hundreds of suppressor tRNAs to alter the rules of the genetic code to enable new research tools and help solve important problems in biotechnology and medicine.

We have included discussion (page 9) on some of the work representing the optimization of the tRNA species for orthogonal translation as the Reviewer suggests. Indeed, the orthogonal translation field, which is more rapidly growing than the utilization of tRNAs to recode nonsense mutations and restore function, has set many milestones in repurposing tRNAs. However, as we highlight in the revised text, the repurposing of tRNAs to incorporate noncanonical amino acid substrates (e.g. orthogonal translation) share some few common principles with the tRNA utilization to correct premature termination codons, but also require many specific aspects in the refactoring. Thus, we felt that reviewing the achievements of orthogonal translation field belongs more to the discussion than to the introduction.

In the introduction (page 2) we concentrated on the examples of repurposing tRNAs to decode nonsense stop codons. We included seminal publications from the field from some of the groups the Reviewer mentioned. However, we concentrated on those that optimize tRNAs and did not review genome's or ribosome repurposing or aminoacyl-tRNA-synthetases modulations as those are aspects unique to the orthogonal translation field. Several seminal reviews, which we cite, represent those developments outside the tRNA refactoring.

2. Outdated view on tRNA/mRNA interactions: What I found particularly frustrating is their statements about the novelty of their structural observations. The fact that suppressor tRNAs are recognized by the ribosome as any other tRNAs is extremely trivial and already known. For instance, Marina Rodnina and Holger Stark observed the very same fact in their cryo-EM structure of E. coli ribosome bound to the suppressor tRNA, tRNA-Sec (<https://pubmed.ncbi.nlm.nih.gov/27842381/>). Having read their introduction, I was expecting to find their engineered tRNAs outperforming all currently existing suppressor tRNAs in genetically-defective human cell lines (e.g. cell lines from beta-thalassemia patients), a test that is commonly used in the field. Instead, I found that the authors used E. coli and existing technologies to confirm existing knowledge. I suggest journals like ACS Chemical Biology as a proper fit for this work.

Since we considered this a recoding rather a classical stop-codon suppression event, we did not include it in the previous version. However, we agree with the Reviewer that this is another example of stop-codon recognition and have now mentioned on page 7 that the conformation of the Sec-tRNA^{Sec} during recoding is indeed similar to that observed here. We have also included an additional image to emphasize this (Fig. 4d). In the discussion (page 8) we compare the features of selenocysteine suppression trio as an evolutionarily selected classical stop-codon suppression system with repurposing tRNAs to use elongation resources and decode stop codons.

Reviewer #3:

The manuscript by Albers and colleagues aims to use tRNA design principles to engineer an improved tRNA that can be used to suppress stop codons. If such a nonsense suppressor tRNA could be identified, it could be useful to suppress premature stop codons that cause human disease. To identify an improved nonsense suppressor tRNA, the authors use the well-defined E. coli translation system. They first choose tRNA(Ala) as a sequence framework, due to the fact that this tRNA has nucleotide identity elements for charging by alanyl-tRNA synthase (AlaRS) that are not located in or near the anticodon. They predominantly reside in the acceptor stem, with one additional nucleotide in the D loop (Fig. 1). The authors first generated 10,000 tRNA variants computationally and narrowed these down to 5 based on predicted secondary and tertiary folding algorithms. These tRNAs, t1-t5, were then transcribed in vitro and tested for aminoacylation. Of these five, t1 was the most efficiently aminoacylated (Fig. 2) and was used for further engineering to improve in vivo nonsense suppression of a GFP reporter. Mutations in t1 in the anticodon stem-loop and the variable arm were ineffective, but mutations that optimized the TphiC stem, with additional D loop modifications (tRNA t1A3DT2), improved nonsense suppression dramatically (Fig. 3C-D). These TphiC stem and D loop mutations were inspired by biochemical insights into how other tRNA bodies are tuned for translation. The authors chose tRNA(Glu) for TphiC mutations and tRNA(Pro) for D loop mutations, due to tight binding of tRNA(Glu) to EF-Tu and interactions of tRNA(Pro) with EF-P when in the P site. The authors further confirmed incorporation of Ala using tandem MS/MS analysis (Fig. S5). The authors then used purified 70S ribosomes loaded with tRNA t1A3T2 to determine a 2.9 Å cryo-EM structure of the ribosome with a stop codon and tRNA t1A3T2 in the A site. The structure shows that the tRNA has the expected conformational features of an elongator tRNA properly bound in the decoding site, with the previously defined rRNA-tRNA contacts.

Overall, this is an interesting paper that will be of wide interest to the translation field. However, there are a few critical holes in the presentation the authors should fill. These are described below.

We were pleased to read that overall the Reviewer finds our results to be of wide interest to the translation field and we thank them for their critical comments.

Major

1. The authors used T7-transcribed tRNAs for all of their biochemical work in vitro. Yet presumably the tRNAs are at least post-transcriptionally modified to some extent in cells. The authors see differences between tRNA behavior in vitro compared to in vivo that could be explained by these differences in modification. For example, nonsense suppression in vivo for t1A3DT2 is much better than t1A3T2 in vivo (Fig. 3C-D), but is worse in the dipeptide assay (Fig. 3E). The authors should map the post-transcriptional modifications of t1A3T2 and t1A3DT2, at least.

The Reviewer raises a very valid and important point that posttranslational modifications could potentially change the effect on suppressor variants in vivo and this could explain variations in activity between *in vivo* and *in vitro* assays as we observed for t1A3DT2. Quantitative mapping of modifications, despite their importance, is so far not possible for all modifications as suggested by the extant literature (PMID: 24625781, PMID: 28488916, PMID: 28488916). Furthermore, the extent of modifications at single nucleotide positions in one tRNA isoacceptor vary largely, e.g. with some nucleotides being uniformly modified in all copies of the same isoacceptor and some nucleotides being modified only in a subset of tRNA isoacceptor copies. This quantification is not trivial and to the best of our knowledge not possible for all modifications. The importance of the modifications as another layer in modulating tRNA stability and activity drives the developments in the field which will hopefully allow in few years tRNA modifications to be quantitatively mapped.

Reflecting on the Reviewer's comment, we included some additional experiments, which indirectly refer to the role of putative modification and explain the observed differences in the suppressor activity *in vitro* and *in vivo*. We included:

- (1) a t1A3D construct which complements the whole spectrum of constructs for the t1 variant and probes the contribution of the D-region alone (Fig. 3)
- (2) another construct based on the t2 tRNA body, in which we also dissect the contribution of each single segment for the tRNA suppression activity (Supplementary Fig. 3)

The readout of this experiment - measuring activity *in vivo* – is indeed an indirect assessment of the contributions of both sequence alterations and posttranscriptional modifications, with no nucleotide resolution, but the exchange of one segment at the time, the contribution of each segment (also modified) can be sampled. The approach also assumes that the modifications in the tRNA segments are independent of changes in distant parts of the tRNA, which is supported by the current literature on (t)RNA modifications suggesting that they depend on local tRNA sequence context, rather than on the whole tRNA sequence. Briefly, we conclude the following (see subsection: "Tuning TΨC- and D-regions markedly enhances stop-codon suppression" page 5 and 6 and revised Fig. 3b and Supplementary Fig. 3):

- (a) Tuning of the TΨC-stem sequence majorly improves the stop codon suppression efficiency.
- (b) Changes in the anticodon loop synergistically with the TΨC-stem modulate suppression activity.
- (c) The D-region exhibits sequence-specific effects which can be neutral to negative. The negative effect can be counteracted by posttranscriptional modification *in vivo* rendering it neutral.

Furthermore, we discuss the role of modifications for eEF1A binding (page 8/9) reviewing the available information from Modomics data base. We thank the Reviewer for raising this point, as we believe that with the inclusion of these additional variants we can confidently emphasize the key regions that need to be modulated to repurpose tRNAs into efficient nonsense suppressors *in vivo*.

2. The authors made acceptor stem changes to t2 (t2AS2 and t2AS3) that made these variants essentially as good as t1A3. (The authors also don't make clear that t2AS3 and t1 have identical acceptor stems, which confused the reader.) Yet they did not test their utility in the GFP nonsense suppression assay in cells. Could these work as well as the t1A3T2 and/or t1A3DT2 variants, i.e. would the TphiC stem loop and D loop variants be required for the t2AS3 family of tRNAs?

We apologize for not being specific about the similarities of the acceptor stem of t1 and t2 families and had relied only on the graphical sequence representation to capture similarities and dissimilarities. In the revision, we explicitly mentioned in the legend of Supplementary Fig. 1 that the successive changes in the t2 acceptor stem to improve charging led to essentially same acceptor stem as t1.

Furthermore, we followed the comment of the Reviewer to test the effect of the D-region and TΨC-stem on the suppression activity of t2 tRNAs. However, we took the t2AS2 variant which still has differences in the acceptor stem compared to t1, unlike t2AS3 which is identical to t1 (Supplementary Fig. 3 and text on p. 5; see also the above comment 1). With these additional experiments we can select key regions to modulate when repurposing tRNAs into efficient nonsense suppressors.

3. The *in vitro* aminoacylation experiments in Fig. 2 are somewhat confusing. The levels of charging for t1 and CUA seem reasonable, but the percentages for t3-t5, GGC and UGC seem

much higher than supported by the gel shown. The gel is a bit smeary, which could lead to overestimation in these samples.

To be able to distinguish the small mass difference between aminoacylated and nonacylated tRNA forms we run very long gels in which the tRNA migration area widens as circles, rendering this method semiquantitative. By representing rounded numbers, we wanted to emphasize on the semiquantitative character on this method. We included a note in the Methods section (page 20) to emphasize the necessity of performing many replicates (see also source Fig. 2) An example of how we performed this quantification is illustrated in Fig. R1. Because of the blurred shape of the bands, we consider the intensity of the whole area: for the amount of aminoacylated tRNA fraction the intensity within the yellow square and for the total amount of tRNA, aminoacylated and non-aminoacylated, the intensity within the red square, respectively.

Figure R1: Example of the performed quantification to estimate the approximate level of aminoacyl-tRNAs. The fraction of aminoacyl-tRNAs (e.g. the intensity of the area within the yellow square) was divided by the total amount of tRNA, aminoacylated and non-aminoacylated, loaded into the lane (e.g. the intensity within the red square).

4. Related to the aminoacylation assays, the authors used a ligation approach to ensure the ACCA was present on the transcribed tRNAs (Fig. S1). However, the ligation is far from quantitative. Could the presence of N+1 (non-templated addition) species be the cause of lower aminoacylation efficiency *in vitro*? Did the authors run a sequencing-grade gel to assess the amount of N+1 tRNA in their preparations? Have the authors tried the approach of Kao et al. (1999) RNA 5, 1268-ff to reduce non-templated species?

The Reviewer is correct in their remark that *in vitro* T7 transcription can yield N+1 extended tRNAs, which would lower the ligation efficiency of the fluorescent oligonucleotide and/or the aminoacylation levels *in vitro*. We use this method only as a comparison and not absolute quantification; the latter is not possible because of the N+1 fragment. N+1 fragments would be present in any *in vitro* synthesized tRNA batch, even in the control (here GGC-tRNA^{Ala}) to which we compare all constructs. This is evident from the non-ligated lower band in all samples to which the oligonucleotide was added (lanes designated with + in the Supplementary Figure 1a lower panel, i.e. stained with SybrGold). Our reasoning is that despite the presence of N+1 product or any other species with nonintact CCA ends, we use this approach as an initial comparative screen for determining the fraction of active tRNAs for each variant, i.e. tRNAs with intact CCA ends. Purposely, we did not quantify the ligated tRNA. Our aim was only to present qualitative evidence that the fraction of active tRNAs, with intact CCA ends, was indeed comparable for all variants to that of the native tRNA^{Ala} (Supplementary Fig. 1a). Hence, we do not believe that the observed differences in aminoacylation (Fig. 2) were due to variations in the fraction of tRNAs with intact CCA ends.

Reflecting on the comment of the Reviewer, we feel that we were unprecise in our description of (i) the selectivity of the approach to only obtain intact CCA ends and (ii) its comparative character. We have therefore included at many places specifications to emphasize this (legend to the suppl. Fig. 1a, Methods section, page 19). We also cite our previous work (PMID: 24009533), which shows that *in vitro* and *in vivo* the fractions of tRNAs with intact CCA ends vary and elaborates that the method itself cannot be used for absolute quantification

within one sample but rather to compare between samples and conditions. Briefly, as shown in Fig. R2, even natural *E. coli* tRNAs, do not display 100% ligation and only a fraction of them have intact CCA ends.

Figure R2: Probing the intact 3'-CCA ends of natural and *in vitro* transcribed tRNAs. The 3'-CCA ends of natural tRNAs from total RNA of *E. coli* cells and *in vitro* transcribed tRNA variants t1, t1A3T2 and Ala(CUA) were probed with a Cy3-labeled hairpin oligonucleotide with a complementary 5'-TGGN-3' overhang (+) and compared to non-ligated tRNAs (-). tRNAs from total RNA were either deacylated prior to probing (+) or not (-). tRNAs were detected by fluorescence (left panel) or stained with SYBR gold (right panel). CUA is tRNA^{Ala}(GGC) with exchanged anticodon to decode UAG stop codon.

5. The FACS bar graphs in Fig. 3C and Fig. S3B should be accompanied by the FACS histograms, to show the cutoff value for GFP fluorescence used for defining GFP levels.

We have included FACS histograms along with all replicates of the immunoblots in the source figures 3d, S2b, (see source data file).

6. Fig. S9. It is not at all obvious that tRNA(Ile) is enriched in this experiment. Is this plot correct? Perhaps simply using the quality of the cryo-EM density of the tRNA and mRNA in the P site, it would be possible to show the tRNA is consistent with tRNA(Ile)?

We agree with the Reviewer that the way the tRNA microarrays were represented, the enrichment of the tRNA^{Ile} was not clearly visible. Thus, we used the cryo-EM density to validate that the P-site tRNA is tRNA^{Ile} since we can model the complete ErmCL nascent chain that is attached to the P-site tRNA. This is now mentioned on page 6 and highlighted in a new series of panels a-c in Supplementary Fig. 9. We have removed the microarray data and associated text to avoid confusion.

7. All of the work was done in *E. coli*, yet the motivation is in part to address human disease. I don't think it's necessary to do more experimental work in this regard, but I do think the Discussion should be expanded to address what's known and what's not known about tRNA functional interactions in the human context. This is especially true when considering the differences in EF-Tu vs. EF1-alpha, and the likely differences in tRNA modifications that would occur in human vs. *E. coli* cells.

We appreciate this suggestion and have included discussion on page 8/9. Briefly, despite lacking quantitative information on thermodynamic contributions of nucleotide pairs to the binding of eEF1A, the homology with EF-Tu including conserved sites to bind aminoacyl-tRNAs, suggest that the optimization principles we established for repurposing bacterial

tRNAs are transferrable to eukaryotic tRNAs. We also acknowledge here that modifications, which in eukaryotic tRNA context are still incompletely elucidated, modulate this binding.

Minor:

1. For comparison purposes, the authors should show the GFP levels from t1A3 in Fig. 3C and 3D, instead of just in Figure S3A.

We have included the expression levels of t1A3 in Fig. 3c.

2. The right half of Fig. S7A seems incorrect. Should this not show purified disomes collapsing to monosomes after RNaseH treatment?

We have corrected this panel.

3. The authors should state that the affinity-purified tRNA for cryo-EM was not aminoacylated when the ribosome complex was formed. This was not clear upon first reading.

We have stated in the Materials and Methods section that the isolated tRNA is not aminoacylated (p.22 and p. 23)

REVIEWERS' COMMENTS

Reviewer #1 (Remarks to the Author):

The authors addressed all my comments.

Reviewer #2 (Remarks to the Author):

Thank you for addressing my comments.

Reviewer #3 (Remarks to the Author):

In the revision by Albers et al., the authors address most of the concerns raised by reviewers 1 and 3. However, reviewer 2 raises some important points that require analysis. Although the tone used by reviewer 2 was (in my mind) a bit harsh, his/her points suggest a path forward that I think could improve the manuscript substantially, to the level needed for publication in Nature Communications. Given the maturity of nonsense suppression for the incorporation of unnatural amino acids in mammalian cells, the authors could rather easily repurpose some of those tools to express their mutant suppressor tRNAs in mammalian cells and test for their function.

As an example, a simplified version of systems used by the Elsässer group could be employed. There are many plasmids available from Addgene (search <https://www.addgene.org/search/catalog/plasmids/?q=elsasser>) that could be used for guidance. For example, many of these use 7SK or U6 promoters to express PyIT tRNAs in human cells. These could be swapped out for the various tRNA(Ala) versions the authors have made, both efficient and inefficient. For a more quantitative assay, the authors should probably use stable transduction rather than transient transfection.

If the authors were to include UGA nonsense suppression data from mammalian cells, properly quantified, then this would overcome the weakness of using in vitro transcribed tRNAs and the E. coli system. Mammalian expression data would also overcome the lack of knowledge about how these tRNAs are post-transcriptionally modified. These experiments would directly relate to the rationale for the study, engineering tRNAs that could eventually be used to treat disease.

Reviewer #1:

The authors addressed all my comments.

We are very grateful to Dr. Amunts for his critical and very helpful comments.

Reviewer #2:

Thank you for addressing my comments.

Again, we thank this Reviewer for their critical comments and the suggestion to include the discussion on repurposing tRNAs to incorporate noncanonical amino acids.

Reviewer #3:

In the revision by Albers et al., the authors address most of the concerns raised by reviewers 1 and 3.

We are pleased to see that the Reviewer feels that we have addressed all their concerns and thank them for their critical and helpful comments.

However, reviewer 2 raises some important points that require analysis. Although the tone used by reviewer 2 was (in my mind) a bit harsh, his/her points suggest a path forward that I think could improve the manuscript substantially, to the level needed for publication in Nature Communications. Given the maturity of nonsense suppression for the incorporation of unnatural amino acids in mammalian cells, the authors could rather easily repurpose some of those tools to express their mutant suppressor tRNAs in mammalian cells and test for their function.

As an example, a simplified version of systems used by the Elsässer group could be employed. There are many plasmids available from Addgene (search <https://www.addgene.org/search/catalog/plasmids/?q=elsasser>) that could be used for guidance. For example, many of these use 7SK or U6 promoters to express PylT tRNAs in human cells. These could be swapped out for the various tRNA(Ala) versions the authors have made, both efficient and inefficient. For a more quantitative assay, the authors should probably use stable transduction rather than transient transfection.

If the authors were to include UGA nonsense suppression data from mammalian cells, properly quantified, then this would overcome the weakness of using in vitro transcribed tRNAs and the E. coli system. Mammalian expression data would also overcome the lack of knowledge about how these tRNAs are post-transcriptionally modified. These experiments would directly relate to the rationale for the study, engineering tRNAs that could eventually be used to treat disease.

In this final assessment the Reviewer cross comments the previous comments of Reviewer #2 and suggests experiments with ectopically expressed tRNAs and using tools developed for non-canonical amino acids incorporation. At this stage, this suggestion is rather surprising for us, particularly in light of the discussion we incorporated in response to Reviewer's #2 comments. As we elaborate in the discussion, the repurposing of tRNAs for correcting nonsense mutations and for non-canonical amino acid incorporation requires fairly distinct strategies. In addition, the suggested plasmid-based strategy to recode nonsense mutations and restore function seems less suitable. While in the synthetic biology – the field with the widest applications of orthogonal translation – a complete manipulation and reprogramming of the genetic activities of the cell (including ectopic plasmid-borne expression and/or incorporation into the genome) is desirable, for therapeutic purposes rather RNA

vaccine-like strategies are favored, i.e. administration of in vitro transcribed tRNAs is preferred (considering the rampant developments of the RNA-based therapeutics and mRNA vaccines, e.g. PMID 33414215; PMID:32728218; PMID: 29567706; PMID: 31342441; PMID: 32893005; PMID: 33816449). Hence, we respectfully disagree with the Reviewer that a replica of experiments developed and optimized for non-canonical amino acid incorporation is easily adapted, and more importantly, not well-suited for stop-codon suppression in the context of nonsense mutation diseases.